# Revisiting the Volume Hypothesis

**Ari Pakman** [1]  **Lior Kreimer** [1]  **Yakir Berchenko** [1]

## Abstract

Modern deep neural networks often contain far more parameters than needed to fit their training data, yet they achieve impressive generalization. A common explanation for this success is the implicit bias of stochastic gradient descent (SGD). An alternative *volume hypothesis* posits that, within low training-loss regions, loss-landscape basins leading to strong generalization occupy much larger regions of weight space than basins that generalize poorly, and therefore SGD is simply more likely to land in the former. Recent experimental explorations of this idea present seemingly contradictory results. While in one set of experiments randomly sampling the network weights until achieving zero training error yielded poor generalization, molecular dynamics density estimates supported the volume hypothesis. We observe that these experiments were performed at different dataset size regimes, and explore an intermediate regime using the Replica Exchange Wang–Landau algorithm to estimate the joint density of states over training and test accuracies in binary networks. Across several architectures and datasets, we show that the generalization advantage of gradient learning over random sampling training generally diminishes as the training data size grows, suggesting a resolution of the paradox.

## 1. Introduction

From the perspective of classical learning theory (Shalev-Shwartz & Ben-David, 2014), it is surprising that modern neural networks generalize so well to new data. These models usually contain many more parameters than are necessary to fit the training set, and making them even larger tends to improve rather than harm generalization (Hestness et al., 2017). In an over-parameterized network, countless parameter configurations can drive the training loss to zero—some generalize, others do not. Yet, for reasons that remain unclear, stochastic gradient descent (SGD) routinely lands on parameter configurations that do generalize (Zhang et al., 2017).

The prevailing view attributes this success to an implicit bias that SGD introduces when it trains an over-parameterized model (Soudry et al., 2018; Gunasekar et al., 2018; Arora et al., 2019; Vardi, 2023). However, a recent work by Chiang et al. (2022) proposed a different explanation, the *volume hypothesis*, according to which within the low training error regions of the weight space, strong generalization regions simply occupy a much larger volume than those that generalize poorly. This idea echoes related proposals in (Pérez et al., 2019; Berchenko, 2024) and has been explored in several theoretical works in simplified cases (Hanin & Zlokapa, 2023; Buzaglo et al., 2024; Harel et al., 2024; Alexander et al., 2025). If correct, one could argue that architectural bias is the main driver of generalization, while SGD's implicit bias is secondary.

This claim has been recently tested through two different approaches, with opposite outcomes. On the one hand, the work by Peleg & Hein (2024) used a *Guess and Check (G&C)* procedure: randomly sampling weight values until achieving zero training error. The resulting networks have worse generalization error than SGD.[1] On the other hand, the work by Yang et al. (2026) uses molecular dynamics techniques to estimate the density of the network parameters as a function of (i) the training loss and (ii) the generalization loss (for regression) or accuracy (for classification). For low training loss, in many classification and regression models the density has a peak at generalization values similar or better than those reached by highly-optimized SGD. This implies the validity of the volume hypothesis, dubbed *high-entropy advantage* by Yang et al. (2026), though in one of their examples the phenomenon disappears for wide enough networks.

---

[1]Department of Industrial Engineering and Management, Ben-Gurion University of the Negev, Beer Sheva, Israel. Correspondence to: Ari Pakman <pakman@bgu.ac.il>.

*Proceedings of the 43rd International Conference on Machine Learning*, Seoul, South Korea. PMLR 306, 2026. Copyright 2026 by the author(s).

[1]Peleg & Hein (2024) argue that earlier claims to the contrary by Chiang et al. (2022) can be explained away by proper initialization and loss normalization.

In order to reconcile these diverging results, we note that the experiments reported in Peleg & Hein (2024) were restricted to binary classification tasks in the low sample regime (up to 32 training samples). Extending G&C to multiclass classification or bigger training datasets is unfeasible due to the high computational cost, which increases quickly with training sample size and the number of classification categories. On the other hand, the experiments in Yang et al. (2026) were performed in the high sample regime. This suggests that the validity of the volume hypothesis might depend strongly on the data size. In this work we explore this idea using the Wang-Landau algorithm (Wang & Landau, 2001), a technique developed in the statistical physics community to compute the probability density of the energy or other macroscopic quantities in systems in which this density can vary across many orders of magnitude. In our case, we use it to estimate the joint density of states $g(A, Q)$ over training accuracy $A$ and test accuracy $Q$ in selected ranges of interest, for multiclass classification networks and training datasets of up to 600 samples. In three different architectures and two datasets, we find the generalization advantage of SGD over random sampling generally diminishes as the training data size grows. This result bridges the different results previously reported. Our results demonstrate a data-dependent transition in which optimization-induced bias dominates in small-data regimes, while architectural volume effects emerge and concentrate with increasing data. This clarifies the respective roles of optimization, architecture, and data in overparameterized generalization.

## 2. Related works

**Wang-Landau in machine learning.** An estimation of the density of states in binary neural networks using the Wang-Landau method was recently performed in (Mele et al., 2025). However, this work only considered the density of the training error, and thus does not yield insights on generalization performance. Another recent use of the Wang-Landau method in machine learning is the work (Liu et al., 2023), which computed the density of output values of a network over the input space, for fixed, previously-trained weights. The volume hypothesis was recently explored by Yang et al. (2026) using a variant of the Wang-Landau method that combines molecular dynamics with non-parametric density estimation. The same technique was recently applied in (Zhang et al., 2025) to the grokking phenomenon.

**Overparameterization and generalization.** The question of why deep neural networks generalize well despite severe overparameterization has been investigated for several decades from a variety of theoretical and empirical perspec-

tives (Wolpert, 1995; Bartlett & Mendelson, 2001; Hoffer et al., 2017; Jakubovitz et al., 2019). Classical learning theory relates generalization to capacity control via complexity measures such as the VC dimension (Vapnik & Chervonenkis, 1971), suggesting that highly expressive models should overfit (Shalev-Shwartz & Ben-David, 2014; Hastie et al., 2001). From this viewpoint, networks with enough parameters to interpolate arbitrary labels would be expected to generalize poorly. Empirically, however, modern neural networks defy this prediction. Even models capable of perfectly fitting the training data often exhibit strong test performance (Haeffele & Vidal, 2017; Nguyen & Hein, 2018), and increasing model size can further improve generalization (Belkin et al., 2019; Neal et al., 2019; Bartlett et al., 2020). A striking demonstration by Zhang et al. (2017) showed that convolutional networks can memorize random labels while still generalizing well on structured data, highlighting a disconnect between expressivity and generalization. Additional work indicates that neural networks tend to learn simple or low-frequency patterns before memorizing noise (Arpit et al., 2017) and that both bias and variance may decrease as model size grows (Neal et al., 2019).

**Implicit bias induced by optimization.** A prominent line of research attributes successful generalization in deep learning to the implicit bias of gradient-based optimization methods (Neyshabur et al., 2015). For linearly separable problems, Soudry et al. (2018) showed that gradient descent converges to maximum-margin solutions, and Arora et al. (2019) argued that the regularization induced by gradient-based training cannot be captured by explicit penalties alone. Subsequent studies have explored how optimization dynamics shape learned representations, including the influence of batch size (Galanti & Poggio, 2022), gradient noise (Liu et al., 2020), effective dimensionality reduction during training (Advani et al., 2020), and simplified dynamics in wide networks (Lee et al., 2020). More recently, Andriushchenko et al. (2023) demonstrated that large learning rates in SGD promote low-rank feature learning.

**Loss landscape and flatness viewpoints.** Another family of approaches links generalization to geometric properties of the loss landscape, particularly notions of flatness or sharpness (Dziugaite & Roy, 2017; Keskar et al., 2017; Jiang et al., 2020; Foret et al., 2021). While such measures are often correlated with generalization, their causal relevance remains debated (Andriushchenko et al., 2023). Related ideas appear in the Bayesian literature, where wide optima are associated with higher posterior mass and improved predictive performance (Izmailov et al., 2018; Wilson & Izmailov, 2020).

**Architectural bias and the volume hypothesis.** Beyond optimization, several works emphasize biases intrinsic to the network architecture itself. Huang et al. (2020) hypothesized that poorly generalizing minima occupy comparatively small regions in parameter space. Building on this intuition, Mingard et al. (2021) argued that, under strong assumptions such as infinite width, SGD behaves similarly to Bayesian sampling, suggesting that architectural bias dominates optimization effects. However, these approximations do not directly apply to finite, practical networks. Other work has questioned the necessity of SGD stochasticity altogether, showing that deterministic gradient descent with explicit regularization can achieve comparable performance (Geiping et al., 2022).

Most directly related to our study, Chiang et al. (2022) proposed the *volume hypothesis*, arguing that generalization is primarily governed by the relative volume of well-generalizing solutions induced by the architecture, with the implicit bias of SGD playing only a secondary role. Our definition of the volume hypothesis states that, among weights with zero training error, the volume of regions with low *generalization* error is big. This differs from the definitions espoused recently in (Scherlis & Belrose, 2025; Fan et al., 2025), which focus on the volume of low *training* error basins.

**Simplicity bias.** A recurring theme in recent work is that modern overparameterized architectures do *not* behave as if they were selecting a hypothesis uniformly at random from an enormous function class; instead, common parameterizations and initialization/training pipelines induce a highly non-uniform *distribution over functions* that is sharply skewed toward simple (structured, compressible, low-complexity) predictors. Pérez et al. (2019) make this point concrete by analyzing the parameter-function map and the induced function-space prior: this map is many-to-one with the probability of realizing a given function varying by orders of magnitude, and empirical as well as theoretical evidence supports an *exponential* relation between function probability and descriptional complexity, yielding an explicit simplicity bias mechanism. Teney et al. (2024) reinforce this picture from an architectural standpoint, showing that "random networks are not random functions": already at initialization, an overwhelming fraction of parameter space corresponds to functions of characteristic (often low) complexity. Mingard et al. (2025) connect these observations to an Occam/algorithmic information theory perspective, by studying the distribution of functions induced by random networks and showing that probabilities can decay approximately exponentially with suitable complexity proxies, while also demonstrating that modifying the regime (e.g., toward "chaotic" behavior) weakens the bias and harms generalization.

In particular, both Berchenko (2024) and Buzaglo et al. (2024) make the Guess-and-Check mechanism explicit and connect it to a classic PAC-style interpretation (Berchenko (2024) studies a "naive algorithm", which is equivalent to Guess-and-Check). The grand picture from both is the following. Let $\hat{p}_S$ denote the success probability of a single guess to come up with the target function. Then the stopping time $T$ is bounded by a geometric random variable with mean $\mathbb{E}[T] = 1/\hat{p}_S$. Under simplicity bias induced by parameter redundancy, "simple" target-functions have comparatively large prior mass, while complex hypotheses have exponentially smaller mass; hence $\hat{p}_S$ can be non-negligible when the target is simple, implying only few trials until interpolation. Consequently, this converts the analysis into a standard finite-class template: the expected number of distinct hypotheses tried until Guess-and-Check stops is on the order of $1/\hat{p}_S$, so one may view the procedure as implicitly selecting from an *effective* hypothesis class of size $|H_{\text{eff}}| \approx 1/\hat{p}_S$, yielding familiar PAC sample-complexity behavior scaling as $\log |H_{\text{eff}}| \approx -\log \hat{p}_S$. In this sense, the simplicity-biased induced prior supplies the analogue of a finite hypothesis class, and generalization bounds follow the same logic as PAC-learning with a finite class—only with complexity controlled by the probability mass assigned by the construction-induced distribution rather than by the raw number of parameters.

In this function-space framing, the so-called *volume hypothesis* (informally: "generalizing solutions occupy large volume") is best viewed as a *consequence* of simplicity bias rather than a competing primitive. Moreover, as a foundational explanation volume is either tautological or ill-defined, with "generalizing well" not an intrinsic attribute of a training set alone (it depends on the out-of-sample distribution/evaluated target), whereas simplicity bias is a well-defined property of the learning setup through its induced probability space over hypotheses.

## 3. Background: the Wang-Landau algorithm

The Wang-Landau algorithm, introduced in (Wang & Landau, 2001), revolutionized Monte Carlo simulations by enabling efficient sampling of the entire energy spectrum of statistical mechanics systems, including rare high-energy states that conventional methods struggle to reach. In statistical mechanics, the density of states $g(E)$ counts how many microstates exist at each energy level $E$. This quantity is fundamental because all thermodynamic quantities can be derived from it (e.g. partition function, internal energy, heat capacity). However, calculating $g(E)$ directly is usually

computationally intractable, because it requires enumerating all possible microstates. In our setting the macroscopic quantities of interest are the train and test accuracies, which we denote as $A$ and $Q$ respectively, for a fixed dataset. We seek to estimate $g(A, Q)$, the volume in the space of network parameters (weights and biases) with given train and test accuracies.

Let us consider a neural network with $N$ binary parameters. The values of $g(A, Q)$ provide a partition of the $2^N$ possible networks, i.e., $2^N = \sum_{A,Q} g(A, Q)$. Computing $g(A, Q)$ via direct counting, i.e. evaluating $A$ and $Q$ for all $2^N$ parameter combinations is infeasible. Instead, the Wang-Landau algorithm typically yields an accurate estimate of $g(A, Q)$. Note that to verify the volume hypothesis we are interested in comparing the value of $Q$ which maximizes $g(A = A_{\max}, Q)$ with the test accuracy obtained from SGD. But in general it might be of interest to estimate the $g(A, Q)$ also for a range of $A \leq A_{\max}$ values.

Assume a prior uniform distribution over the $N$ binary parameters of the network $\mathbf{w} \in \{\pm 1\}^N$. If we knew the value of $g(A, Q)$, reweighting this distribution with a factor $1/g(A, Q)$ would yield a uniform distribution on the $(A, Q)$ space. Of course $g(A, Q)$ is unknown. The key insight of the Wang-Landau algorithm is to perform a random walk in the $\mathbf{w}$ space reweighted by $1/g(A, Q)$, while continuously updating the current estimate of $g(A, Q)$ until we reach a flat histogram in the $(A, Q)$ space. The update to $g(A, Q)$ consists in simply multiplying by a factor $f > 1$, $g(A, Q) \leftarrow g(A, Q)f$, to discourage future visits to the current state $(A, Q)$.

At each step, the algorithm maintains (i) $\log g(A, Q)$: the current estimate of the log-density of states, (ii) $f$: the modification factor (typically initialized at $e$), and (iii) $H(A, Q)$: a histogram that counts visits to each pair of train/test errors, from which a criterion to occasionally diminish $f$ is obtained. After initializing $g(A, Q) = 1$ (up to an overall normalizing factor), $H(A, Q) = 0$, $f = e \simeq 2.718$, each step of the random walk consists of

1. Propose a move in the space of parameters (e.g., flip a random subset of binary parameters), $\mathbf{w} \to \mathbf{w}'$.

2. Calculate the accuracies $(A' = A'(\mathbf{w}'), Q' = Q'(\mathbf{w}'))$ associated with the proposed state.

3. Accept with probability
$$P = \min(1, g(A, Q)/g(A', Q')). \quad (3.1)$$
If accepted, set $\mathbf{w} \leftarrow \mathbf{w}', A \leftarrow A', Q \leftarrow Q'$. Note that (3.1) coincides with the Metropolis acceptance rate for a target distribution proportional to $1/g(A, Q)$.

4. Update histogram: $H(A, Q) \leftarrow H(A, Q) + 1$. Update the log-density: $\log g(A, Q) \leftarrow \log g(A, Q) + \log f$.

Along the random walk, we periodically check whether the histogram $H$ is close to being "flat", for example by checking the condition

$$\min_{A,Q} H(A, Q) > z \times \text{mean}_{A,Q}\left[H(A, Q)\right], \quad (3.2)$$

with typical $z \geq 0.8$. When the flatness condition (3.2) is satisfied, the histogram is reset to $H(A, Q) = 0$, the modification factor reduced as $f \leftarrow \sqrt{f}$ and the random walk continues. Convergence is reached when $f < f_{final}$ for some small $f_{final}$. Several parameters can be tuned for optimal speed, such as the size of the bins for $A, Q$, the number of spins $s$ flipped in the proposal moves and other schedules for the factor $f$.

### 3.1. Parallelization via Replica Exchanges

While the Wang-Landau algorithm is ergodic and provably converges (Jacob & Ryder, 2014), for models with large ranges of $(A, Q)$ values it may require extremely long runs. An effective solution, inspired by the parallel tempering method (Geyer et al., 1991), is to split the accuracy space $(A, Q)$ into several small overlapping regions, each running an independent Wang-Landau random walk (Vogel et al., 2013; 2014). Periodically, a proposal is made for overlapping regions to exchange configurations, and it is accepted with a standard Metropolis-Hastings probability for an interchange proposal. This approach is called Replica Exchange Wang-Landau (REWL), and we adopted it in our experiments.

## 4. Experiments

We performed experiments using three different architectures detailed in Table 1, on the MNIST and Fashion-MNIST datasets. For each dataset and architectures we considered three train data sizes $D = 30, 300, 600$. In all the cases, we considered balanced training sets with equal sizes for all ten categories.

We implemented the REWL algorithm with four and six random walkers running on workstations with four and six NVIDIA RTX 600 ADA GPUs, in each case assigning one random walker per GPU. The use of the latter is crucial for speed, since for each proposed state $\mathbf{w}'$ the accuracies over the full train and test sets must be evaluated. For the transition proposals $\mathbf{w} \to \mathbf{w}'$, we tuned during initial runs the number of binary weights $p$ to be flipped to obtain accep-

*Table 1.* Network architectures and parameter counts.

**Base Model**

| Layer | Input → Output | Kernel/Stride /Padding | Params |
|---|---|---|---|
| Conv2D | $1 \times 28 \times 28 \to 6 \times 28 \times 28$ | 3/1/1 | 54 |
| ReLU | | | |
| MaxPool | $6 \times 28 \times 28 \to 6 \times 14 \times 14$ | 2/2/0 | |
| FC1 no bias | $1176 \to 64$ | | 75,264 |
| ReLU | | | |
| FC2 no bias | $64 \to 10$ | | 640 |
| Total | | | 75,958 |

**Deeper Model**

| Layer | Input → Output | Kernel/Stride /Padding | Params |
|---|---|---|---|
| Conv2D | $1 \times 28 \times 28 \to 6 \times 28 \times 28$ | 3/1/1 | 54 |
| ReLU | | | |
| MaxPool | $6 \times 28 \times 28 \to 6 \times 14 \times 14$ | 2/2/0 | |
| Conv2D | $6 \times 28 \times 28 \to 6 \times 28 \times 28$ | 3/1/1 | 324 |
| ReLU | | | |
| FC1 no bias | $1176 \to 64$ | | 75,264 |
| ReLU | | | |
| FC2 no bias | $64 \to 10$ | | 640 |
| Total | | | 76,282 |

**Wider Model**

| Layer | Input → Output | Kernel/Stride /Padding | Params |
|---|---|---|---|
| Conv2D | $1 \times 28 \times 28 \to 6 \times 28 \times 28$ | 3/1/1 | 54 |
| ReLU | | | |
| MaxPool | $6 \times 28 \times 28 \to 6 \times 14 \times 14$ | 2/2/0 | |
| FC1 no bias | $1176 \to 128$ | | 150,528 |
| ReLU | | | |
| FC2 no bias | $128 \to 10$ | | 1280 |
| Total | | | 151,862 |

tance rates (3.1) of around $40\%$. This typically resulted in $3 \leq p \leq 9$. We assumed convergence when the modification factor of all the random walkers reached $\log f \leq 2^{-17}$.

**Restricting the estimation range.** For computational tractability, we restricted the ranges of $A$ and $Q$ over which we estimated $\log g(A, Q)$. For the training accuracy $A$ we assumed maximum granularity, and computed $g(A, Q)$ over six values $A \in \{D - 5, D - 4, \ldots, D\}$, where $A = D$ corresponds to all $D$ training points correctly predicted, $A = D - 1$ to all except one, etc. For the test accuracy we aggregated the values into bins containing ten values of the accuracy count. Thus for data sizes $D \geq 30$ the full test sets had $10,000$ samples and $Q$ takes about 1000 values in its full range. The actual range of $Q$ was restricted in order to capture a width of 10% accuracy while including

the location of maximum density, as found by trial-and-error initial estimates. Proposals $\mathbf{w} \to \mathbf{w}'$ that lead to $(A', Q')$ pairs outside the designated range are rejected (but $H$ and $\log g$ are still updated, as with any rejected proposal).

# 5. Results

The results of all the estimated density curves for interpolation solutions, $\log g(A = D, Q)$, are presented in Figure 3. Note the rapid decay of the probabilities as $Q$ moves away from the maximum. Thus random sampling training would land with high probability on $Q^* = \arg\max_Q g(A = D, Q)$.

**Diminishing advantage of gradient descent.** The generalization accuracies $Q^*$ for all the cases are presented in Table 2, where they are compared with results from (stochastic) gradient descent learning. This table shows one of our central observations: in most cases, the advantage of gradient-based training diminishes as the data size grows, thus reconciling the observations of (Peleg & Hein, 2024) and (Yang et al., 2026) into a unifying data-size dependent framework. Figure 1 presents the generalization accuracy gap between random sampling and gradient learning in all the cases.

**Sharpening of the density curves.** Another important result is illustrated in Figure 4: across all architectures and datasets, the curvature of the generalization accuracy density sharpens with increasing dataset size. This is consistent with the idea that the probability volume of well generalizing solutions shrinks as the data size grows, and with recent results on the data-size dependence of the volume of low training-loss basins (Fan et al., 2025).

**The role of width and depth.** We observe in Figure 3 a consistent increase of generalization accuracy for wider networks. This differs from the results in (Peleg & Hein, 2024), where network width improves SGD results but not those of Guess-and-Check. We also observe mixed effects of deeper networks, whereas in (Peleg & Hein, 2024) the effect of depth is reported to be negative overall. These discrepancies seem to be a function of the particular networks or datasets chosen and do not seem to be central to the analysis of the volume hypothesis.

**Variability of estimates.** Figure 2 illustrates the typical variability of the REWL density estimates across different runs of the algorithm.

*Table 2.* Generalization accuracy (%) of deep binary networks trained via random sampling vs. (stochastic) gradient descent (GD/SGD). Random sampling training corresponds to the maximum of the density curves in Figure 3. GD and SGD training were performed using the Binary Connect algorithm (Courbariaux et al., 2015) with the Adam optimizer (learning rate $10^{-4}$). SGD was performed with mini-batches of size $5, 30, 60$ for datasets of sizes $D = 30, 300, 600$ respectively, balanced over ten classification categories.

| | | | **Dataset: MNIST** | | | |
|---|---|---|---|---|---|---|
| Train Size | Model | Random | GD | Gap | SGD | Gap |
| | Base | 28.5 | $58.7 \pm 3.5$ | 30.2 | $55.9 \pm 5.3$ | 27.4 |
| 30 | Deeper | 29.5 | $55.2 \pm 4.2$ | 25.7 | $46.2 \pm 10.1$ | 16.7 |
| | Wider | 30.9 | $59.5 \pm 3.8$ | 28.6 | $58.0 \pm 7.8$ | 27.1 |
| | Base | 66.6 | $83.2 \pm 0.8$ | 16.6 | $78.3 \pm 8.0$ | 11.7 |
| 300 | Deeper | 70.5 | $78.8 \pm 5.8$ | 8.3 | $77.0 \pm 5.1$ | 6.5 |
| | Wider | 68.7 | $84.1 \pm 0.9$ | 15.4 | $83.6 \pm 2.9$ | 14.9 |
| | Base | 75.0 | $88.5 \pm 0.5$ | 13.5 | $81.7 \pm 4.1$ | 6.7 |
| 600 | Deeper | 72.9 | $85.5 \pm 4.3$ | 12.6 | $80.0 \pm 5.3$ | 7.1 |
| | Wider | 77.1 | $89.0 \pm 0.5$ | 11.9 | $87.3 \pm 3.9$ | 10.2 |

| | | | **Dataset: Fashion-MNIST** | | | |
|---|---|---|---|---|---|---|
| Train Size | Model | Random | GD | Gap | SGD | Gap |
| | Base | 31.7 | $58.8 \pm 3.1$ | 27.1 | $56.3 \pm 2.7$ | 24.6 |
| 30 | Deeper | 32.4 | $53.6 \pm 4.2$ | 21.2 | $49.9 \pm 3.9$ | 17.5 |
| | Wider | 33.4 | $61.2 \pm 1.2$ | 27.8 | $59.0 \pm 1.2$ | 25.6 |
| | Base | 62.1 | $76.5 \pm 1.6$ | 14.4 | $69.4 \pm 5.6$ | 7.3 |
| 300 | Deeper | 61.5 | $72.8 \pm 1.1$ | 11.3 | $67.6 \pm 6.6$ | 6.1 |
| | Wider | 62.0 | $77.5 \pm 0.9$ | 15.5 | $76.9 \pm 1.6$ | 14.9 |
| | Base | 67.3 | $76.4 \pm 4.0$ | 9.1 | $75.7 \pm 2.5$ | 8.4 |
| 600 | Deeper | 65.6 | $71.0 \pm 5.2$ | 5.4 | $72.4 \pm 3.7$ | 6.8 |
| | Wider | 68.1 | $79.6 \pm 0.4$ | 11.5 | $79.8 \pm 0.5$ | 11.7 |

# 6. Discussion

Our results provide a unified explanation for recent contradictory results regarding the volume hypothesis and the role of optimization in generalization. By explicitly probing intermediate training set sizes, we show that the relationship between random sampling and gradient-based training is strongly data-dependent. When training data are scarce, interpolating solutions are abundant but highly heterogeneous in test performance. In this regime, SGD consistently reaches atypical regions of parameter space that generalize substantially better than typical random interpolating solutions. As the dataset grows, however, the density of interpolating states becomes increasingly concentrated around a narrow range of test accuracies. In this regime, the typical interpolating solution approaches the performance achieved by SGD, and the apparent advantage of optimization diminishes.

These findings clarify the interpretation of the volume hypothesis. Rather than a universal explanation independent of optimization, volume effects emerge progressively as data

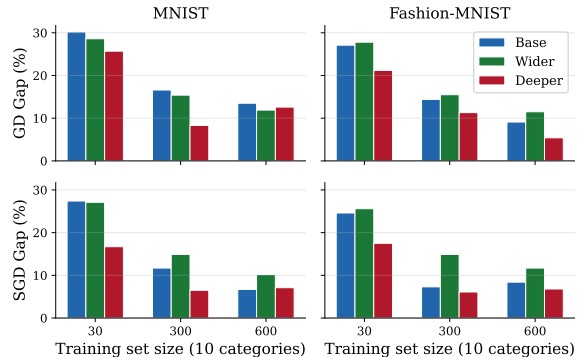

*Figure 1.* Generalization gap of GD/SGD over random sampling.

constraints increase. Optimization-induced bias is therefore essential in small-data regimes, while architectural bias increasingly shapes the geometry of solution space as more data are observed. From this perspective, architectural and optimization biases are complementary mechanisms whose relative importance depends on dataset size.

Our results are also consistent with recent work on simplicity bias in overparameterized models. The observed sharpening of density curves with increasing data can be interpreted as a concentration of probability mass onto a restricted subset of functions compatible with the training set. In this sense, volume effects reflect a macroscopic consequence of architectural simplicity bias under growing constraints, rather than an independent primitive.

Finally, we emphasize that our study focuses on binary-weight networks and moderate-scale architectures, which enable explicit density estimation but differ from typical continuous-weight models. Our goal is not to directly model modern large-scale training, but to isolate fundamental geometric effects that are otherwise difficult to observe. Extending density-of-states methods to broader settings remains an important direction for future work.

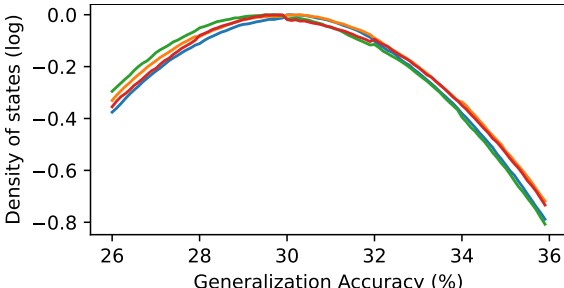

*Figure 2.* **Variability of the Wang-Landau density estimator**. Generalization accuracy density from four independent runs of the REWL algorithm on MNIST with $D = 30$, using the Deeper model.

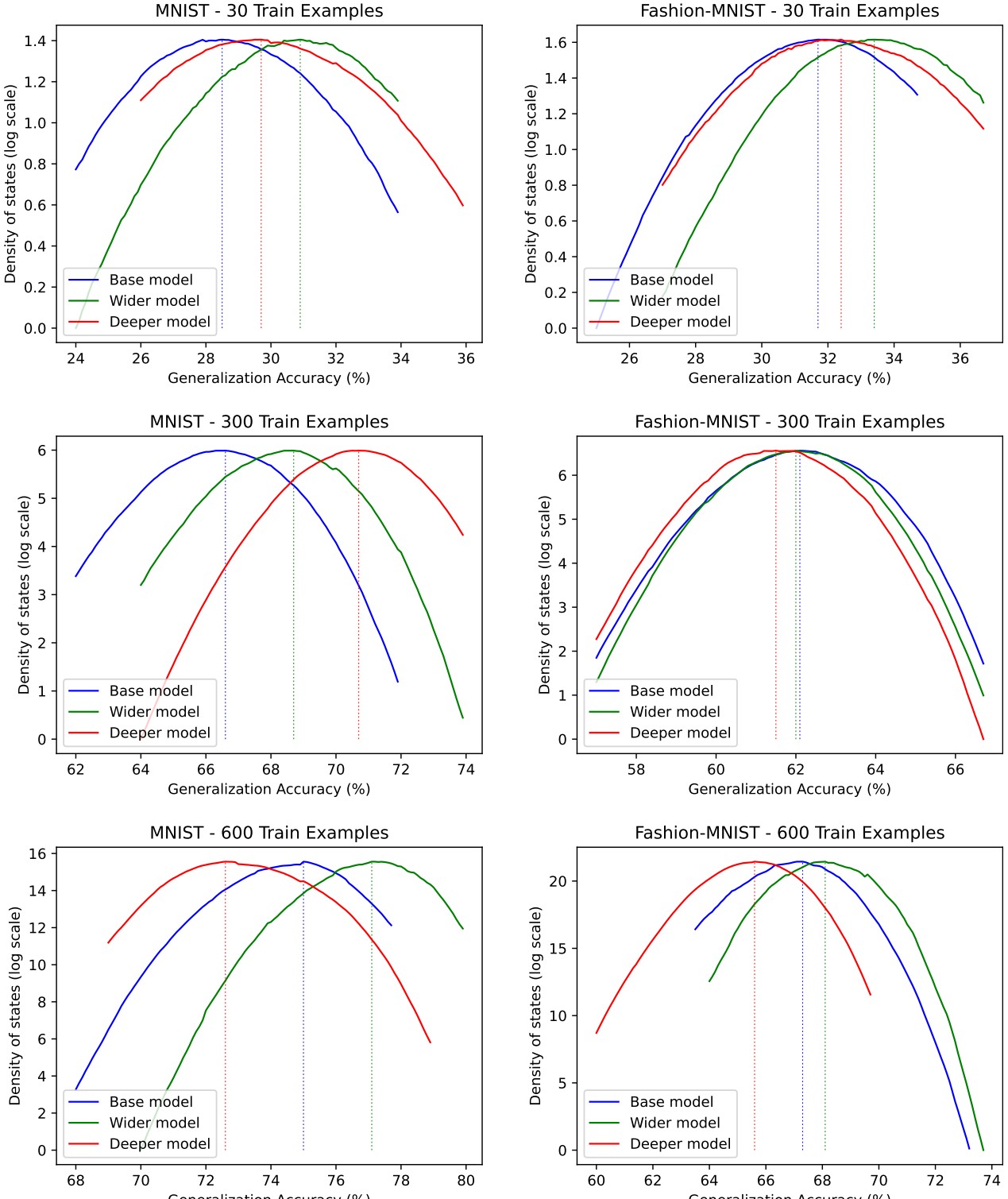

*Figure 3.* **Density of interpolating states as a function of the generalization accuracy.** For each dataset we show log-density curves $\log g(A = D, Q)$, for the three binary network models detailed in Table 1. The log-densities are defined up to an additive constant. The vertical dotted lines indicate maximum density locations. All the configurations had zero training error on datasets with balanced labels.

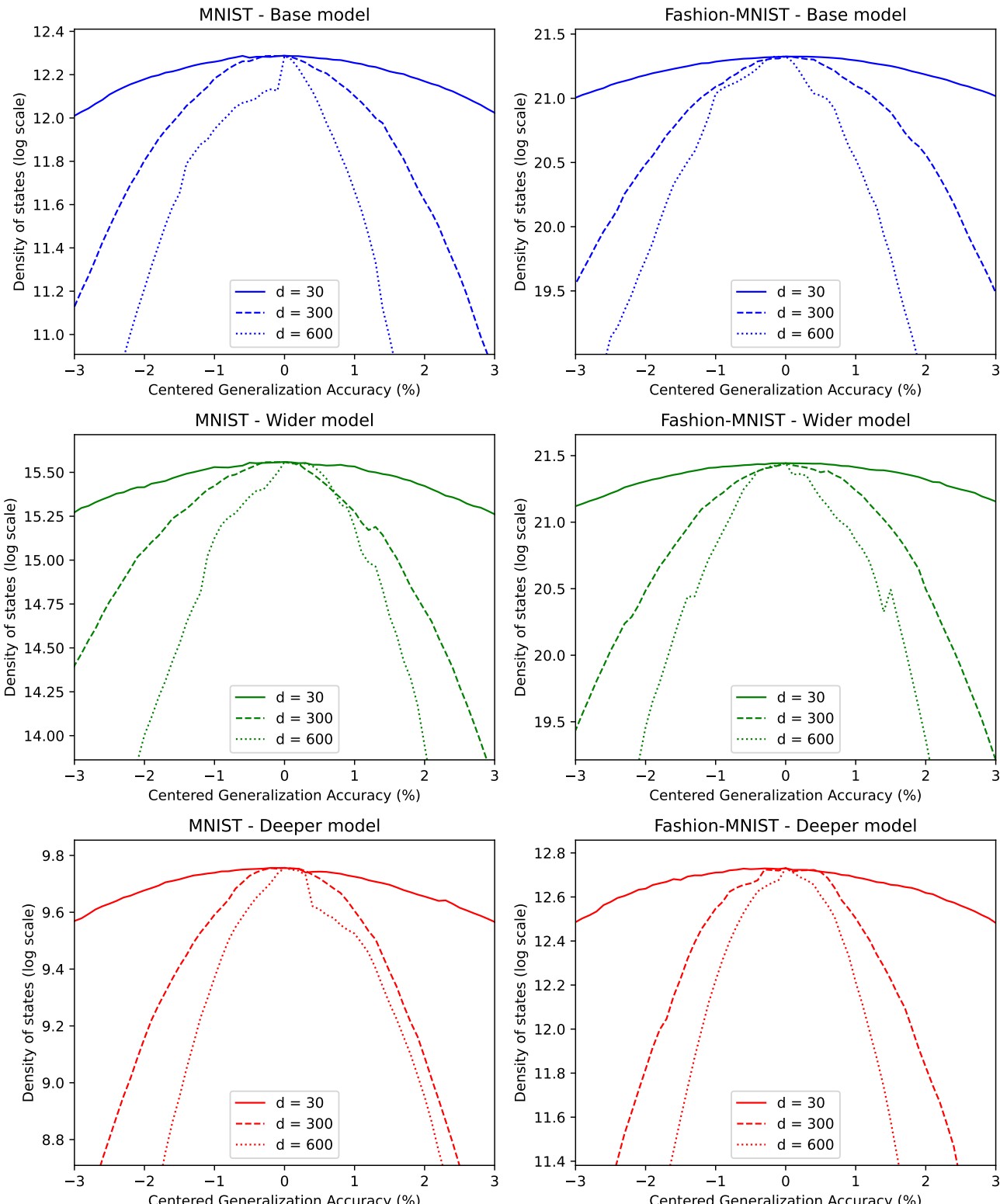

*Figure 4.* **Curvatures increase with dataset size.** These curves are a close-up of those in Figure 3, but here each panel shows density curves for the same model and dataset, varying the dataset size. The curves were vertically and horizontally shifted to have their maxima at the same location. Note that in all cases the curvatures increase with the size of the training dataset.

## Acknowledgements

A.P. is supported by the Israel Science Foundation (grant No. 1138/23) and by the Israel Ministry of Innovation, Science and Technology (Israel-France collaboration 2025-2028).

## Impact Statement

This paper presents work whose goal is to advance the field of machine learning. There are many potential societal consequences of our work, none of which we feel need to be specifically highlighted here.

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

*Table 3.* Iterations and running time for the models with $D = 300$ and $D = 600$ data points.

| Model | Iterations per Walker | Wall clock time (hours) |
|---|---|---|
| MNIST 300 Base model | 412,240,000 | 274 |
| MNIST 300 Deeper model | 461,880,000 | 307 |
| MNIST 300 Wider model | 599,880,000 | 399 |
| Fashion-MNIST 300 Base model | 466,520,000 | 311 |
| Fashion-MNIST 300 Deeper model | 624,880,000 | 416 |
| Fashion-MNIST 300 Wider model | 647,560,000 | 431 |
| MNIST 600 Base model | 456,560,000 | 304 |
| MNIST 600 Deeper model | 753,880,000 | 502 |
| MNIST 600 Wider model | 968,600,000 | 645 |
| Fashion-MNIST 600 Base model | 549,080,000 | 366 |
| Fashion-MNIST 600 Deeper model | 681,120,000 | 454 |
| Fashion-MNIST 600 Wider model | 910,280,000 | 606 |

## A. More details on the Replica Exchange Wang Landau algorithm

### A.1. Running times

In our implementation, each random walker in the REWL algorithm executed about $1.5 \times 10^6$ iterations per hour. Table 3 indicates the number of iterations and wall clock time for the more demanding models until convergence ($\log f \leq 2^{-17}$). Note that models with more parameters or more training data take longer to converge, in some cases taking up to three weeks.

### A.2. Random-walkers aggregation.

To aggregate the log densities $\log g(A, Q)$ estimated by different random walkers, we added to each density a different constant in order to minimize the squared differences between pairs of $\log g(A, Q)$ estimated in overlapping regions. This freedom follows from the fact that $g(A, Q)$ is defined up to an overall normalization constant. The resulting combined curve is illustrated in Figure 5. The final log-density curve is obtained by the mean of the $\log g(A, Q)$'s in overlapping regions.

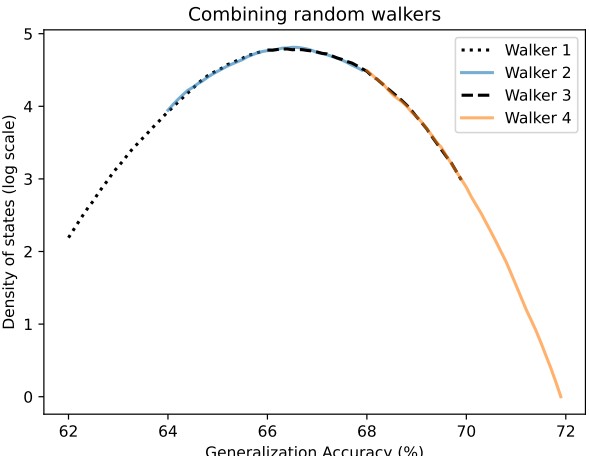

*Figure 5.* **Combining different random walkers' results.** Estimated $\log g(A = D, Q)$ from four random walkers restricted to the ranges $[62.0, 65.9], [64.0, 67.9], [66.0, 69.9]$ and $[68.0, 71.9]$, for the Base Model on MNIST with $D = 300$ training samples. The curves are vertically rigidly displaced to minimize the squared distance of the overlaps.

