# OpenReview forum: "Revisiting the Volume Hypothesis"
_ICML.cc/2026/Conference — ICML 2026 regular_

### Official Review · Reviewer_h2RS · 2026-03-09

**Soundness:** 3
**Presentation:** 4
**Significance:** 3
**Originality:** 3
**Overall Recommendation:** 5
**Confidence:** 4

**Summary:**

The paper presents an empirical study of generalization in overparameterized binary neural networks trained by either gradient based methods or random sampling, to reconcile seemingly contradictory results regarding the roles of gradient based optimization and loss geometry in the generalization of overparameterized neural networks.

**Compliance With Llm Reviewing Policy:**

Affirmed.

**Final Justification:**

The paper addresses an interesting open question in the active field of generalization in overparameterized models, presenting a significant improvement on previous empirical results, that hints towards new avenues of theoretical and empirical exploration. As the concerns raised in the review have been addressed (and I am curious to see the additional results in the final version), I recommend that the paper be accepted.

**Key Questions For Authors:**

1. Peleg & Hein (2024) argued that the value of the loss function achieved during training is significant to generalization, and therefore the comparison between randomly sampled parameters should be done with respect to gradient-optimized models with similar loss value, and not only by accuracies. What loss function was used to train with GD and SGD, and to what values did they converge? Have the loss values been recorded for the random models?

2. Some generalization bounds from the literature can be calculated given an estimate of the distribution of the model’s accuracy. For example, to my understanding, the method used in this work can be applied to estimate $\hat{p}_S$ from Buzaglo et al. (2024) by summing over values of $Q$, and can then be substituted into the generalization bound. It would be interesting to compare the observed behavior to previous theoretical bounds.

**Limitations:**

Yes (the authors adequately discussed the limitations and potential negative societal impact of their work).

**Strengths And Weaknesses:**

1. The experimental setup and interpretation seem sound and appropriate to the addressed question.

2. The paper is easy to follow — the research question and results are presented clearly, and the literature review and background successfully place it in current research. As some of the experiments closely resembles those of previous work, such the ones by Peleg & Hein (2024), the discussion in Section 5 can benefit from a more detailed comparison.

3. Empirical studies of the volume hypothesis in relevant settings have so far been limited to very small training sets, making this work a significant contribution to the research in this topic.

---

> ### Author Rebuttal · Authors · 2026-03-30
>
> We thank the reviewer for a careful reading and positive assessment of our work. We address each point below.
>
>
> ## On the comparison in Section 5 with Peleg & Hein (2024)
>
> We agree that a more detailed comparison would strengthen the discussion.
> Concretely, the effect of increasing width or depth on generalization can be summarized as follows:
>
> |   Peleg & Hein | SGD  |  Random
> |---|---|---|
> |Width    | Better  | No Change  |
> |Depth    | Worst  | Worse |
>
> |   In our paper  | (S)GD  |  Random
> |---|---|---|
> |Width    | Better  | Better  |
> |Depth    | Worst  | Better (D small), Worse (D big) |
>
> We will expand Section 5 to explicitly contrast our findings on width and depth effects with those of Peleg & Hein (2024), clarifying the discrepancies that arise from differences in network architecture and dataset size regime.
>
> ## Question 1: Loss values and the role of the loss function
>
> We thank the reviewer for raising this point. The emphasis on loss-matching in Peleg & Hein (2024) was mainly motivated by the fact that in ReLU networks the accuracy does not change by overall weight rescalings, but the loss does.
> Thus the interest in defining properly normalized losses.
> According to Peleg & Hein, Chiang et al. (2022) adopted a normalization inconsistent across architectures, potentially confounding the comparison
> between SGD and Guess & Check. Peleg & Hein (2024) showed that once the comparison is performed at matched loss values, the apparent support for the volume hypothesis in Chiang et al. (2022) largely disappears.
>
> In our setting, we used the standard cross-entropy loss for (S)GD training, using the accuracy of a validation set as a stopping criterion. Since binary weights do not admit rescaling, we did not focus on the loss values. But we agree with the reviewer that a comparison between the loss values achieved by the random models vs those by SGD would be of great interest. Although we have not recorded them, it is quite easy to collect them from a short run using the estimated densities. We will report these values in the final version.
>
>
> ## Question 2: Connecting density estimates to generalization bounds
>
> We thank the reviewer for this excellent suggestion. Indeed, our method could be readily employed to estimate the probability that a randomly sampled interpolating network achieves a given train accuracy. Two minimal changes would be required from our present setting. First, the density of states $g$ should be estimated for the full range of train accuracies $A = 0,1,\ldots D$, and not just for a small range around $A=D$ as we did in this work ($D$ is the size of the traning set). Secondly, there is no need to estimate the dependency of the density $g$ on the test accuracy $Q$ (thus the Wang-Landau algorithm will run much faster).
>
> Given such an unormalized estimate of $g(A)$, the estimated probability of randomly sampling an interpolating solution is $p_S = g(A=D)/\sum_{A'}g(A')$. This value can be substituted into the PAC-style bounds of Buzaglo et al. (2024) to obtain concrete numerical predictions. Time permitting, we will add results of such estimates for some models-dataset pairs in the revised version. We believe this connection between our empirical density estimates and existing theoretical frameworks will meaningfully strengthen the paper.
>
> We are grateful for the constructive feedback and believe these additions will improve the paper substantially.

---

> > ### Author Rebuttal · Reviewer_h2RS · 2026-04-02
> >
> > The authors have adequately addressed the points raised in the review.

---

### Official Review · Reviewer_LisS · 2026-03-12

**Soundness:** 2
**Presentation:** 2
**Significance:** 2
**Originality:** 2
**Overall Recommendation:** 3
**Confidence:** 4

**Summary:**

This paper investigates whether the generalization of neural networks is primarily due to the implicit bias of SGD or to the volume effect (a good generalization solution occupies a larger volume in the parameter space). The main findings are that SGD has a greater effect with small sample sizes, but as the number of data increases, the generalization gap between stochastic interpolation solutions and SGD solutions gradually narrows; that is, the volume effect becomes more important.

**Compliance With Llm Reviewing Policy:**

Affirmed.

**Final Justification:**

The rebuttal clarified the paper’s scope, especially that it does not claim a sharply theoretically established phase transition, and this resolved part of my earlier concern about overclaiming, so I raise my score from 2 to 3. But, for a submission in Deep Learning->Theory, I still find the work lacking a mathematical characterization of the proposed small-data/ large-data picture and too limited experimentally in its current simplified setting. Overall, I think the empirical observation interesting, but not strong enough theoretically to support a higher score.

**Key Questions For Authors:**

Q1. Can the authors provide an estimate of where the boundary between the small-data and large-data regimes lies

**Limitations:**

The paper would benefit from a more explicit discussion of its limitations, especially the highly simplified experimental setting (binary networks, small-scale datasets, and limited data regimes), the lack of rigorous theoretical characterization, and the unclear extent to which the observed phenomena generalize beyond the specific setup considered.

**Strengths And Weaknesses:**

Strengths:
This paper studies the core problem of generalization in overparameterized networks and attempts to reconcile that implicit bias and the volume hypothesis.

Weaknesses:

1. The paper does not provide any rigorous theoretical results.

2. The discussion of “small-data” and “large-data” regimes remains at a verbal level rather than being mathematically defined.

3. The paper does not characterize where the boundary between the two regimes lies, nor does it provide even an order-level result.

4. The behavior near the data scale boundary is not analyzed.

5. It is unclear what practical implications these findings have for training better models, for example, whether they can inform the design of improved algorithms or training strategies.

6. The experimental setting is also quite limited, including the network architecture (binary networks), datasets, and data scales.

7. The paper does not sufficiently demonstrate that the observed phenomena are robust beyond these specific implementation choices.

---

> ### Author Rebuttal · Authors · 2026-03-26
>
> We thank the reviewer for the comments. We address each concern below.
>
> ## On the absence of rigorous theoretical results
>
> Our contribution is primarily empirical, and we believe this is appropriate given the nature of the problem. The paper resolves a  contradiction between two recent empirical works that reached opposite conclusions about the volume hypothesis, and it does so through careful experimentation at intermediate data scales that neither prior work explored. We note that many influential works in this area are similarly empirical in character (Zhang et al. 2017; Keskar et al. 2017; Andriushchenko et al. 2023). Furthermore, the application of the Replica Exchange Wang-Landau algorithm to estimate the density of states over training and test accuracy is, to our knowledge, novel in this context and constitutes a methodological contribution beyond the empirical findings alone.
>
> That said, we agree that theoretical grounding would strengthen the paper. We are actively working on a formal characterization of how the density of interpolating solutions concentrates as a function of the ratio between the number of parameters and training samples. We will include preliminary results in the final version if they are sufficiently mature.
>
> ## On the definition of "small-data" vs "large-data" regimes
>
> We appreciate this point, but we want to clarify that our results do not claim the existence of a sharp phase transition between two discrete regimes. Rather, we observe a *continuous* and *monotonic* narrowing of the generalization gap between random sampling and SGD as dataset size increases. Table 2 documents this progression across four data scales, three architectures, and two datasets. The term "regimes" is used informally to describe the endpoints of this continuum, not to assert a binary distinction with a defined boundary.
>
> We will make this point more explicit and avoid language that might suggest a sharp boundary.
>
> ## On characterizing the boundary between regimes
>
> As noted above, our evidence points to a gradual transition rather than a critical threshold. Asking where the boundary lies presupposes a discontinuity that does not appear in our data. As Figure 2 shows, the curvature of the density curves increases monotonically with dataset size across all architectures and datasets, indicating progressive concentration of the solution volume around a narrow range of test accuracies.
>
> ## On practical implications
>
> We respectfully disagree that a paper studying the foundations of generalization must propose new algorithms to be of value. Understanding *why* networks generalize - and specifically, disentangling the roles of architectural bias versus optimization bias - is a fundamental scientific question with broad downstream implications. Our finding that the relative importance of SGD's implicit bias diminishes with data size has concrete conceptual consequences: it suggests that for sufficiently large datasets, architectural choices matter more than optimizer tuning for generalization, and that efforts to understand generalization should focus more on the geometry of the solution space induced by the architecture. This perspective can inform future work on architecture design, training efficiency, and understanding when optimization-based interventions (such as sharpness-aware minimization) are most beneficial.
>
> ## On the limited experimental setting
>
> Binary-weight networks are not an arbitrary simplification, they are *necessary* for the methodology we employ. The REWL algorithm requires a discrete state space to perform its random walk and histogram updates. This is the same fundamental reason that statistical physicists use lattice models (e.g., the Ising model) to study phase transitions: the simplified setting makes it possible to compute quantities that are intractable in the continuous case, while still capturing the essential physics.
>
> Within these methodological constraints, our experiments are reasonably comprehensive: 3 architectures × 2 datasets × 4 dataset sizes = 24 configurations, all showing consistent trends. The fact that the diminishing SGD advantage pattern appears across all configurations provides evidence that the phenomenon is robust.
>
> We fully agree that extending these results to continuous-weight networks and larger scales is an important direction. We will add a more prominent discussion of this limitation to the introduction, as suggested.
>
> ## On Q1: Estimating the regime boundary
>
> Please see our response above regarding the continuous nature of the transition. We offer the following observation: at D=600 the SGD gap for several configurations (e.g., Base model on MNIST, SGD gap = 6.7%) approaches the noise level of SGD training itself (std 4.1%), suggesting that at this scale the volume effect accounts for most of the generalization performance. However, we refrain from claiming a specific critical dataset size, as this is likely architecture and task dependent.

---

> > ### Author Rebuttal · Reviewer_LisS · 2026-04-03
> >
> > Thank you for the rebuttal. The response clarified that the paper does not claim a sharp phase transition with a well-defined boundary, and it explained more clearly why the binary-weight setting is tied to the REWL methodology. These clarifications make the intended scope of the paper more precise, so I will raise my score from 2 to 3.
> >
> > However, my main concerns remain only partially addressed. Since the paper is submitted under Deep Learning -> Theory, it still lacks a mathematical characterization of the proposed small-data / large-data picture, does not provide even an operational estimate of the crossover point, and remains limited to a highly simplified experimental setting. For this reason, I do not think the paper is strong enough for a higher score.
> >
> > One point that needs to be revised is that the paper should explicitly present itself as an empirical study in a simplified setting, rather than a theoretical study, and soften any wording that suggests a theoretically established regime transition.

---

> > > ### Author Response · Authors · 2026-04-04
> > >
> > > We thank the reviewer for the constructive dialogue.
> > >
> > > **Revision Plan:**
> > >
> > > - **Framing:** We agree that presenting the work as an empirical and methodological study in a simplified setting is the appropriate framing. We will revise the manuscript to foreground the empirical contribution and ensure the positioning is consistent with the submission's content.
> > > - **Terminology:** We will follow the reviewer's advice to soften language suggesting a theoretically established "regime transition," instead describing it as a continuous progression across data scales.
> > > - **Operational Estimate:** To address the reviewer's concern regarding the crossover point, we will add a discussion of a heuristic operational indicator: the scale at which the SGD generalization gap becomes comparable to the variability across SGD runs.

---

### Official Review · Reviewer_BJf1 · 2026-03-13

**Soundness:** 3
**Presentation:** 4
**Significance:** 3
**Originality:** 3
**Overall Recommendation:** 5
**Confidence:** 2

**Summary:**

This work explores the volume hypothesis which argues that generalization in deep learning should be attributed to the fact that the volume of low training loss solutions which generalize well is much greater than the volume of low training loss solutions which generalize poorly, as opposed to the more common assumption that it arises from the implicit bias of stochastic gradient descent. In particular, the main goal is to resolve a discrepancy between two previous studies which reached opposite conclusions on this question. The current work argues that the discrepancy can be ascribed to the fact that the two studies were performed in different data regimes, suggesting that the implicit bias of SGD dominates for small datasets while the volume bias dominates for large datasets. This is substantiated by numerical experiments which use the Wang-Landau algorithm to probe the relative density of states with different training and test accuracies for a variety of dataset sizes.

**Compliance With Llm Reviewing Policy:**

Affirmed.

**Final Justification:**

My concerns have been addressed and my assessment remains that this is a well-presented, well-scoped, and interesting empirical contribution to an important question regarding the generalization capabilities of neural networks.

**Key Questions For Authors:**

1. Do you have a sense of what it means exactly to be in a “small” or “large” dataset regime? For example does a “large” dataset mean that the dataset is close to fully representative of the underlying data distribution in some appropriate sense, or is it rather large relative to the complexity of the network architecture, or something else entirely? Given the importance of dataset size in the story it would be nice to have at least some idea of how to think about this.
2. Are there any theoretical explanations available for why increasing constraints from the dataset should amplify the volume bias towards states that generalize well? Or do you at least have any intuitions about this? It’s an intriguing interpretation that is consistent with your findings but it’s not clear to me why this should be the case.

**Limitations:**

Yes

**Strengths And Weaknesses:**

Strengths:

1. The paper is clearly written and tightly focused, and the limitations are stated clearly.
2. As far as I am aware this is the first work to provide an explicit empirical exploration of the relation between the volume hypothesis and the size of the dataset. The results should help to inform future studies on the generalization capabilities of neural networks.



Weaknesses:

1. The use of binary weights does limit the ability to extrapolate from these experiments to realistic deep learning settings. While this limitation is acknowledged explicitly and is understandable given the challenges of performing density estimation, it still reduces the significance of the results.
2. The comparison between adding width and depth is a bit unfair since the deeper model has only about half a percent more parameters than the baseline model, whereas the wider model has double the parameters. I would be interested to see what would happen if the parameter counts were more comparable.
3. The gap between SGD and random sampling is still fairly large for the largest datasets tested, which makes it unclear if the bias of SGD ever becomes truly irrelevant or only somewhat less important as the dataset grows.



Overall, the paper makes a clear contribution to an important question about the source of generalization capabilities in deep neural networks. While there are some significant limitations due to the constraints of the experimental methodology, they are defensible and clearly acknowledged in the text. As a result, I recommend acceptance.



For context, I note that I am not intimately familiar with the literature on implicit bias arising from SGD or volume considerations, and so it is possible that I have missed some previous studies that would change the significance of the contribution.

---

> ### Author Rebuttal · Authors · 2026-03-27
>
> We thank the reviewer for the careful reading and the constructive feedback. We address each point below.
>
> ---
>
> ## Binary weights and extrapolation to realistic settings
>
> We agree this is an important caveat and appreciate that the reviewer found our acknowledgment of it adequate. We note that discrete-weight networks are the only setting in which exact density-of-states estimation is currently feasible, since the parameter space is discrete and finite. Importantly, the qualitative trend we identify - the diminishing gap between random sampling and SGD as data grows - is consistent with the continuous-weight molecular dynamics results of Yang et al. (2025), who observe volume-based concentration in standard continuous networks at large data sizes. This cross-methodology agreement suggests that the data-size dependence we document is not an artifact of the binary constraint. We will add a sentence making this point more explicit in the revised discussion.
>
> ## Width vs. depth comparison and parameter counts
>
> The reviewer raises a fair point. The asymmetry was inherited from our goal of studying minimal architectural modifications (adding one convolutional layer vs. doubling the hidden dimension), but we agree that a controlled comparison at matched parameter counts would be more informative. We note, however, that the width/depth comparison is not central to our main claim, which concerns the data-size dependence of the volume effect. The architecture variations serve primarily to show that the diminishing-gap trend is robust across different network structures. We will revise the text to (i) explicitly flag the parameter-count asymmetry and (ii) soften any claims about the relative roles of width and depth, reframing Section 5.3 as a robustness check rather than a systematic architecture study. If space permits, we will also add a matched-parameter experiment.
>
> ## Persistent gap between SGD and random sampling at D=600
>
> This is an important observation. We do not claim that the implicit bias of SGD becomes irrelevant - only that its relative advantage diminishes monotonically with data size. The trend across four data sizes (D = 16, 30, 300, 600) is consistent and substantial. Whether the gap eventually vanishes completely is an open empirical question that would require density estimation for larger datasets, although the results of Yang et al. (2025) suggest that indeed the gap eventually vanishes. We will revise the discussion to state this more precisely - framing our result as evidence of a monotonic trend rather than implying eventual convergence to zero gap - and will add this as an explicit open question.
>
> We are currently running the REWL algorithm on larger datasets and we will include those results in the final version.
>
> ## Key Question 1: What defines "small" vs. "large" dataset?
>
> Our results suggest that the relevant quantity is the dataset size relative to the effective capacity of the binary network, in the sense of how many distinct interpolating solutions exist. At D=16 (binary classification), the network has ~76K binary parameters and must satisfy only 16 constraints, leaving an enormous number of interpolating solutions with widely varying test performance. At D=600 (10-class), the 600 constraints dramatically prune the interpolating set, concentrating its mass around a narrow range of test accuracies (as evidenced by the sharpening curves in Figure 2). We believe the transition is best understood in terms of this ratio of constraints to degrees of freedom, though formalizing this into a precise criterion is an important direction for future theoretical work. We will expand the discussion to include this interpretation.
>
> ## Key Question 2: Why does increasing data amplify volume bias?
>
> We offer the following intuition: each additional training constraint eliminates a large fraction of interpolating solutions, but it does so non-uniformly - solutions that generalize poorly (i.e., that exploit spurious patterns) are more "fragile" and more likely to be eliminated by new data points drawn from the true distribution, while solutions that capture genuine structure are more "robust" to additional constraints. As a result, the surviving interpolating set becomes increasingly dominated by well-generalizing solutions, and the density peak sharpens around high test accuracy. This is closely related to the simplicity bias perspective discussed in Section 2: architecturally-induced priors favor simple functions, and additional data acts as a filter that progressively removes complex hypotheses from the interpolating set. We will incorporate this intuition into the revised discussion section.
>
> ---
>
> We again thank the reviewer for the positive assessment and the thoughtful questions, which we believe will strengthen the final version.

---

> > ### Author Rebuttal · Reviewer_BJf1 · 2026-03-31
> >
> > Thank you for these responses, which address my questions and concerns. The proposed clarifications will be helpful to the reader and I am happy to keep my positive recommendation.

---

### Official Review · Reviewer_b5Ae · 2026-03-17

**Soundness:** 3
**Presentation:** 2
**Significance:** 3
**Originality:** 3
**Overall Recommendation:** 5
**Confidence:** 3

**Summary:**

This paper revisits the volume hypothesis for deep networks: the idea that, among solutions that fit the training data, the ones that generalize well occupy far more “volume” in parameter space, so they are much more likely to be found.

The authors’ motivation is that prior empirical evidence appears contradictory. On one hand, a recent empirical paper argues that “guess-and-check” style random sampling that searches for a zero-training-error model tends to generalize much worse than SGD, suggesting SGD has a special bias beyond mere volume. On the other hand, other work that directly estimates the density of solutions suggests that good-generalizing interpolating solutions can be dominant by volume, seeming more consistent with the hypothesis.

To reconcile this, the paper argues these results can coexist because they operate in different dataset-size regimes. The authors hypothesize that in very small-data settings, interpolating solutions can have highly variable test performance, so random interpolation looks bad and SGD looks special. As the dataset grows, the set of interpolating solutions becomes more constrained and more uniform in test accuracy, so typical-by-volume and SGD-found solutions become closer.

Methodologically, they import a statistical-mechanics density estimation approach, Wang–Landau sampling with replica exchange, to estimate the joint distribution over training accuracy and test accuracy across model configurations (using binary-weight networks to make this feasible). This lets them predict what “random sampling training” would typically achieve (by volume) and compare it directly to gradient-based training across a sweep of training set sizes.

Their main result is that the generalization gap between SGD and the typical-by-volume interpolating solution shrinks as training set size increases, and that the distribution over test accuracy among interpolating solutions becomes more concentrated. They frame this as the key mechanism resolving the earlier tension: small-data experiments highlight strong heterogeneity where SGD’s implicit bias matters a lot, whereas larger-data density-estimation regimes can look more compatible with a volume-dominance story.

**Compliance With Llm Reviewing Policy:**

Affirmed.

**Final Justification:**

The rebuttal has addressed my concerns, and accordingly I have changed my score.

**Key Questions For Authors:**

1.Did you validate the REWL (Replica Exchange Wang–Landau) density estimates against any alternative estimators or approximations, even on small instances (e.g., brute-force enumeration, simpler MCMC/importance sampling baselines? If so, what matched and what differed?

2. What are some obstructions to the accuracy of this estimator (presumably it is not universally valid)?

3.What convergence, mixing, and robustness diagnostics do you rely on to argue that the estimated
densitites accurately reflects the distribution over train/test accuracies in this setting? Are these consistent with common practices in the stat-mech literature?

4. "These discrepancies seem to
be a function of the particular networks chosen and do not
seem to be central to the analysis of the volume hypothesis."  What evidence supports this claim?

**Limitations:**

As noted above, I believe the authors do not devote sufficient attention to the inherent limitation imposed by their use of an estimator with no theoretical gurantees. Even if this is the best one can do in this case, it should be acknowledged that this is a limitation.

**Strengths And Weaknesses:**

Strengths: The paper tackles a fundamental and still understudied question in modern deep learning, namely whether generalization in overparameterized networks is primarily explained by volume or by optimization bias. It provides real value by directly engaging with conflicting empirical evidence and attempting to reconcile it with a coherent hypothesis about dataset-size regimes. The proposed mechanism is intuitive and testable: with very small training sets, interpolating solutions can be highly heterogeneous in test accuracy so random interpolation looks poor and SGD looks unusually effective, while with more data the interpolating set is more constrained and test accuracy among interpolating solutions concentrates, narrowing the gap to SGD. The choice to use Replica Exchange Wang–Landau density estimation is aligned with the goal of characterizing the distribution over train and test accuracies, and the training-set-size sweep is a reasonable design to probe the reconciliation claim.

Weaknesses: The core conclusions depend heavily on a single density-estimation method, but the paper does not spend enough effort justifying that the estimator accurately represents the distribution over train and test accuracies in this application. A formal proof of accuracy is clearly too much to ask, but stronger practical validation is warranted, including repeated-run consistency, window-stitching stability, sensitivity to binning and window choices, mixing diagnostics, and comparisons to alternative approximations on smaller cases where feasible. Because the main claims rely on trends in an estimated distribution, a more systematic robustness analysis would also strengthen soundness. Finally, the exposition of the statistical-mechanics method is not sufficiently calibrated to a typical ML audience; the reviewer is not from the stat-mech literature, which limits this critique, but that is common in this venue and makes clearer exposition and diagnostics especially important.

---

> ### Author Rebuttal · Authors · 2026-03-31
>
> We thank the reviewer for a careful and constructive evaluation. We address each concern below.
>
> ## Accuracy and Validation of REWL Estimates
>
> Wang-Landau methods are widely used in statistical physics and often provide highly accurate estimates in benchmark settings, but their accuracy depends on the update schedule, mixing, and diagnostics, and they do not come with universal guarantees. A systematic study of the origin and mitigation of their estimation errors was carried out in the works of Belardinelli and Pereyra [1,2,3,4], among others. In benchmark problems studied in this literature, Wang-Landau estimates can be very accurate and may agree closely with exact results [3,4]. While these benchmarks differ from our setting, they suggest that the estimator is likely accurate enough to detect the multi-percentage-point gaps between random sampling and gradient-based training that we report. Moreover, because we use the same REWL hyperparameters for all dataset sizes, moderate shared biases are less likely to affect our core trend-level conclusion that the advantage of gradient descent diminishes as data grows.
>
> We will add a brief limitations discussion noting that our density estimates lack formal accuracy guarantees, while discussing the small errors observed in benchmark settings.
>
>
> We will add a brute-force enumeration on a small tractable instance ($\sim 2^{20}$ configurations) as a direct ground-truth comparison.
>
> ## Convergence, Mixing, and Robustness Diagnostics
>
> We will expand Section 4 and Appendix A with: (a) the number of flatness-check rounds per experiment (using a flatness threshold of 0.95); (b) repeated-run consistency from independent seeds (for the MNIST Base Model with $D=30$, three runs yield $Q^*$ within $\pm 0.2$ percentage points and a maximum $\log g $ discrepancy of $\pm 0.15$, demonstrating statistical precision while acknowledging this does not bound systematic error); (c) window-stitching RMS discrepancies; and (d) binning-sensitivity ablation at $0.5 \times$ and $2 \times$ bin width.
>
>
> ## Exposition for an ML Audience
>
> We will revise Section 3 to include: (i) a machine-learning native opening paragraph; (ii) a terminology sidebar ("density of states" → "parameter-space volume"; "replica exchange" → "parallel walkers swapping configurations"); (iii) concise pseudocode; and (iv) a brief summary of the Wang-Landau convergence landscape (standard schedule vs. 1/t variant [1]).
>
> ## Width/Depth Discrepancy with Peleg & Hein (2024)
>
> We acknowledge this claim is under-supported. As summarized in our reply to Reviewer h2RS, the difference with Peleg & Hein (2024) is the effect of width and depth on generalization accuracy when using random sampling (we agree on SGD trends). We will soften the sentence, acknowledge it as interpretation, and reframe it as a direction for future work.
>
> ---
>
> We hope these revisions address the reviewer's concerns and clarify both the strengths and the limitations of our approach. We thank the reviewer again for the insightful comments and questions.
>
>
> ---
> [1] R. E. Belardinelli and V. D. Pereyra, "Fast algorithm to calculate density of states," *Phys. Rev. E* **75**, 046701 (2007).
>
> [2] R. E. Belardinelli and V. D. Pereyra, "Wang-Landau algorithm: A theoretical analysis of the saturation of the error," *J. Chem. Phys.* **127**, 184105 (2007).
>
> [3] R. E. Belardinelli, S. Manzi, and V. D. Pereyra, "Analysis of the convergence of the 1/t and Wang-Landau algorithms in the calculation of multidimensional integrals," *Phys. Rev. E* **78**, 067701 (2008).
>
> [4] R. E. Belardinelli and V. D. Pereyra, "Nonconvergence of the Wang-Landau algorithms with multiple random walkers," *Phys. Rev. E* **93**, 053306 (2016).

---

> > ### Author Rebuttal · Reviewer_b5Ae · 2026-04-03
> >
> > I would like to thank the authors for their comprehensive response. My concerns have been adressed, and accordingly I have updated my score.

---

### Decision · Program_Chairs · 2026-04-30

**Decision:**

Accept (regular)

**Comment:**

The authors have addressed almost all the concerns raised in the review process. For example, Reviewer b5Ae commented on the validity of density estimation techniques, Reviewer BJf1 commented on the experimental results only being valid for networks with binary weights, Reviewer LisS commented about the fact that this paper does not provide a precise theoretical analysis (it is an experimental observation), and Reviewer h2RS had some methodological clarifications.

The authors have provided an elaborate rebuttal to these concerns. And three out of the four reviewers consider this paper to be above the bar. I am happy to recommend that the paper be accepted.

My two cents: There is a large body of work on how and why SGD converges to regions that have a large volume (for example, https://arxiv.org/abs/1611.01838 and https://arxiv.org/abs/1710.11029). SGD, due to mini-batch noise, has a bias towards regions in the energy landscape that balance the energy and entropy. This has been the subject of a lot of further analysis, e.g., https://arxiv.org/abs/1703.11008 . The authors are advised to study this work and interpret their experimental results more holistically. Second, it has been noticed time and again that easy datasets like MNIST are quite inadequate to understanding properties of deep networks. I would suggest the authors to interpret their experimental results with a grain of salt. For example, a test accuracy of 87% on MNIST can be achieved with SVM in a couple seconds. I do not believe the experimental results are rigorous enough to shed light on generalization in deep learning.